# A Novel Fusion Method for State-of-Charge Estimation of Lithium-Ion Batteries Based on Improved Genetic Algorithm BP and Adaptive Extended Kalman Filter

**DOI:** 10.3390/s23125457

**Published:** 2023-06-09

**Authors:** Liling Cao, Changfu Shao, Zheng Zhang, Shouqi Cao

**Affiliations:** College of Engineering Science and Technology, Shanghai Ocean University, Shanghai 201306, China

**Keywords:** lithium-ion batteries, state of charge, FFRLS, IGA-BPNN, AEKF, fusion modeling

## Abstract

The lithium-ion battery is the power source of an electric vehicle, so it is of great significance to estimate the state of charge (SOC) of lithium-ion batteries accurately to ensure vehicle safety. To improve the accuracy of the parameters of the equivalent circuit model for batteries, a second-order RC model for ternary Li-ion batteries is established, and the model parameters are identified online based on the forgetting factor recursive least squares (FFRLS) estimator. To improve the accuracy of SOC estimation, a novel fusion method, IGA-BP-AEKF, is proposed. Firstly, an adaptive extended Kalman filter (AEKF) is used to predict the SOC. Then, an optimization method for BP neural networks (BPNNs) based on an improved genetic algorithm (IGA) is proposed, in which pertinent parameters affecting AEKF estimation are utilized for BPNN training. Furthermore, a method with evaluation error compensation for AEKF based on such a trained BPNN is proposed to enhance SOC evaluation precision. The excellent accuracy and stability of the suggested method are confirmed by the experimental data under FUDS working conditions, which indicates that the proposed IGA-BP-EKF algorithm is superior, with the highest error of 0.0119, MAE of 0.0083, and RMSE of 0.0088.

## 1. Introduction

Due to increased environmental requirements and the expansion of the lithium-ion battery industry across the world, there is greater demand on the market for completely electric vehicles [1]. Lithium-ion batteries in the automotive industry are quite common due to their high degree of safety, high power density, and dependable charging and discharging [2]. The most important type of data to monitor in moving vehicles with new energy is the state of charge (SOC) of their batteries. In the battery management system, with accurate SOC estimation, the issue of the overcharging and over-discharging of batteries could be avoided and the life of batteries could be extended through battery equalization, which could ensure the normal and safe functioning of new energy vehicles under demanding operating circumstances [3]. The SOC, however, cannot currently be acquired directly due to the complexity of the internal chemical processes in lithium-ion batteries and the limits of the available technology, and can only be inferred from the performance of the battery’s exterior characteristics [4,5].

At present, there are many kinds of major methods applied to detecting SOC via in-depth studies, including the coulomb counting approach [3,6,7], open circuit voltage method [8,9], equivalent circuit model method [10,11,12,13,14,15], data-driven method [16,17,18,19], and fusion method [20,21,22,23,24,25,26,27,28,29]. The coulomb counting technique is straightforward, but susceptible to current accuracy and cumulative errors. In the open circuit voltage (OCV) approach, SOC is estimated based on the one-to-one relationship between OCV and SOC; however, it necessitates a lengthy rest period for the lithium-ion battery before measuring OCV. Therefore, it is not applicable to moving automobiles.

Currently, the Kalman filter (KF) and its variants are widely used for vehicle state estimation with multi-information fusion. The authors of [30] proposed a velocity-based Kalman filter to estimate the velocity errors, attitude errors, and gyro bias errors of R-INS. The authors of [31] proposed a novel algorithm called YOLOv5-tassel for detecting tassels in UAV-based RGB images. The testing results upon improving the YOLOv5-tassel method achieved a mAP value of 44.7%. In [32], a novel vehicle sideslip angle estimation algorithm with delay compensation was designed by building a vehicle dynamic model and a visual geometry model, and the efficacy of the algorithm was demonstrated via gyratory testing. The authors of [33] used the Kalman filter to estimate yaw misalignment and velocity errors. In [34], the square root cubic Kalman filter was designed to estimate the roll and pitch, to reject the gravity component caused by the vehicle’s roll and pitch when estimating the vehicle slip angle. In [35], to improve the accuracy of independent vehicle positioning under highly dynamic driving conditions during GNSS outages, the adaptive Kalman filter was used to fuse information such as attitude, velocity, and position. In view of these, the KF and its variants have been revealed to have particular advantages in state estimation, and have also undergone a lot of research in the field of SOC estimation for new energy vehicle lithium batteries. The equivalent circuit model method is mainly combined with the H-infinity filtering and KF [13,14] to predict SOC. The authors of [10] introduced three new matrix decomposition algorithms (singular value decomposition (SVD), UR decomposition, and LU decomposition) to solve the divergence problem of Cubature Kalman filtering (CKF), respectively, and conducted a comparison study. The authors of [12] used SVD instead of Cholesky decomposition to improve the Unscented Kalman filter (UKF) and introduced the Sage–Husa adaptive filter to reduce the influence of interference noise on SOC estimation. The authors of [11] used a dual adaptive extended Kalman filter to reduce the effect of inaccurate initial capacity values on SOC estimation.

With the in-depth research on artificial intelligence, the data-driven method of SOC prediction has been largely verified. This data-driven method does not require an equivalent model of the lithium-ion battery, and the SOC is predicted by training the training set based on the mapping relationships between the obtained input variables, such as current, voltage, temperature, charge/discharge rate, and SOC. The authors of [16] used a Long Short-Term Memory network to continuously correct the actual capacity of a lithium-ion battery, so that the prediction error of the optimized coulomb counting method was less than 10%. The authors of [18] used a gated recurrent unit recurrent neural network to estimate the SOC, which avoided the effect of oscillation of the weight change on the prediction results. The authors of [19] trained a deep feedforward neural network (DNN) using battery discharge data at different temperatures and showed that the average absolute error of the validation set at 25 degrees Celsius was 1.10%, with higher prediction accuracy.

Meanwhile, fusion methods for the SOC estimation of lithium-ion batteries have been further investigated. The authors of [25] used the support vector machine method to compensate for the CKF error. In [28], SOC estimation under different temperatures and operating conditions was implemented using a fusion algorithm based on UKF filtering and EKF filtering. The authors of [29] used a genetic algorithm (GA) to optimize the initial parameters of a BP neural network (BPNN) for SOC estimation, but the GA suffered from the problem of premature convergence to a local optimum solution. In [20,21,23,27], the authors used BPNN to compensate for the estimation error of EKF, but the uncertainty in the initial BPNN parameter configuration led to inconsistent prediction results each time. Considering this, in order to improve the estimation accuracy of a battery’s SOC, a novel fusion method for SOC estimation of lithium-ion batteries based on improved genetic algorithm BP and adaptive extended Kalman filter is proposed in this paper.

The main contributions of this paper are as follows:

1. A second-order RC equivalent circuit model is built, the SOC-OCV relationship is obtained, and parameter identification is implemented using the forgetting factor recursive least squares (FFRLS) method.

2. In view of the fact that the EKF algorithm cannot cope with real-time changes in noise during SOC prediction, an adaptive open-window estimation approach is introduced to further reduce the noise interference.

3. A genetic algorithm is optimized by introducing a cosine function to build adaptive crossover probability and variation probability, and the annealing technique to enhance the ability to jump out of local extremes. The improved GA solves the problem of easily maturing in the early stages and stagnating in the later stages, and thus avoids falling into local optimal solutions.

4. The improved GA is used to optimize the initial parameters of the BP neural network (BPNN) that is used to compensate for the estimation error of AEKF. The proposed estimation strategy is applied to numerical simulation experiments, and BP-AEKF, GA-BP-AEKF, and IGA-BP-AEKF are compared. The latter has better performance in accuracy and robustness.

## 2. Model Building and Parameter Identification

### 2.1. Battery Modeling

The precision of the parameter identification of the lithium-ion battery model has a significant bearing on the accuracy of SOC calculation via the equivalent circuit approach [36]. The Thevenin model [37], second-order RC model [38], and Rint model [39] are now the most popular models. The accuracy of the model is enhanced with increasing RC order, but its computational complexity also rises. A second-order RC model is developed in this paper with a detailed analysis, as seen in Figure 1. This model performs better dynamically than the first-order model because it splits the cell polarization into electrochemical polarization and concentration polarization.

As shown in the figure, $U_{OC}$ is the open circuit voltage; $U_{L}$ is the terminal voltage; I represents the battery current, taking the discharge direction as the positive direction; $R_{0}$ is the battery’s internal resistance in ohms; $R_{1}$ and $R_{2}$ represent electrochemical and concentration polarization resistance, respectively; $C_{1}$ and $C_{2}$, respectively, represent electrochemical polarization capacitance and concentration polarization capacitance; and $U_{1}$ and $U_{2}$ are the voltages at the ends of $R_{1}$ and $R_{2}$, respectively.

The function relation of the equivalent circuit model is shown as follows:(1)$$\{\begin{array}{c} \frac{dU_{1}}{dt}=-\frac{U_{1}}{R_{1}C_{1}}+\frac{I}{C_{1}}\\ \frac{dU_{2}}{dt}=-\frac{U_{2}}{R_{2}C_{2}}+\frac{I}{C_{2}}\\ U_{L}=U_{OC}-IR_{0}-U_{1}-U_{2} \end{array}$$

According to [16], the SOC of a lithium battery can be expressed as:(2)$$SOC=SOC_{0}-\frac{1}{Q_{N}}\int \eta Idt$$

The state space in Expression (3) is obtained after the discretization of Equations (1) and (2).
(3)$$\{\begin{array}{c} \left[\begin{array}{c} SOC(k+1)\\ U_{1}(k+1)\\ U_{2}(k+1) \end{array}\right]=\left[\begin{array}{ccc} 1 & 0 & 0\\ 0 & e^{-\frac{\Delta t}{\tau_{1}}} & 0\\ 0 & 0 & e^{-\frac{\Delta t}{\tau_{2}}} \end{array}\right]\left[\begin{array}{c} SOC(k)\\ U_{1}(k)\\ U_{2}(k) \end{array}\right]+\left[\begin{array}{c} -\frac{\eta \Delta t}{Q_{N}}\\ R_{1}\left(1-e^{-\frac{\Delta t}{\tau_{1}}}\right)\\ R_{2}\left(1-e^{-\frac{\Delta t}{\tau_{2}}}\right) \end{array}\right]\\ U_{L}(k+1)=\left[\begin{array}{ccc} \frac{\partial U_{OC}}{\partial SOC} & -1 & -1 \end{array}\right]\left[\begin{array}{c} SOC(k)\\ U_{1}(k)\\ U_{2}(k) \end{array}\right]-R_{0}I(k) \end{array}I_{k}$$
where $SOC_{0}$ is the initial value of SOC; $\tau_{1}$ and $\tau_{2}$ are time constants; $\tau_{1}=R_{1}C_{1}$; $\tau_{2}=R_{2}C_{2}$; Δt is the sampling period; $Q_{N}$ is the rated capacity of the cell; η is the coulombic efficiency; and $\frac{\partial U_{OC}}{\partial SOC}$ was obtained using Equation (4) of the OCV-SOC fitting relationship.

### 2.2. OCV-SOC Acquisition

It was demonstrated that the open circuit voltage and SOC have the same trend and show specific correspondence with the SOC values [21]. After the OCV-SOC fitting relationship is established, $U_{OC}$ can be obtained from the SOC predicted by the AEKF algorithm, and then, used for model parameter identification. In this paper, experiments were conducted with LiMn_2_O_4_ lithium batteries with a rated capacity of 35 Ah, a charging cutoff voltage of 4.2 V, a discharging cutoff voltage of 2.6 V, and a rated voltage of 3.7 V. The OCV-SOC curves were obtained as shown in Figure 2. The authors of [39] investigated the effects of temperature, data points, and aging degree when obtaining the OCV-SOC relationship for Li-ion batteries, and when 11 data points were selected to model the OCV-SOC relationship in the OCV test data, the highest power of the polynomial fit function should not exceed an order of 10. When the highest power of the polynomial is larger, an overfitting phenomenon will occur; when the highest power of the polynomial is smaller, an underfitting phenomenon will occur, so the polynomial with the highest power of seven is selected as the fitting function under comprehensive consideration. The fitting results are shown in Figure 3.

The fitted polynomial equation of the seventh order is:(4)$$\begin{array}{l} U_{oc}(SOC)=39.61SOC^{7}-153.5SOC^{6}+248.8SOC^{5}-221.7SOC^{4}\\ \;+119.3SOC^{3}-39.2SOC^{2}+8.141SOC+2.697 \end{array}$$

### 2.3. FFRLS-Based Parameter Identification

In the field of system identification, recursive least squares (RLS) has been widely used because of its advantages such as ease of understanding, good convergence effect, and fast speed [13]. However, RLS has the problems of data saturation and data accumulation, which affect the accuracy of current parameter identification; the introduction of forgetting factors can effectively prevent data saturation and enhance the accuracy of identification with new data [40]. The specific formula of the forgetting factor recursive least squares method is derived from [40], and the detailed flow of the algorithm is shown in Figure 4.

In the figure, $\theta_{k}$ is the estimated vector of the discriminated parameters at the previous moment; K is the gain matrix; P is the error covariance matrix; $\varphi_{k}$ is the observed data matrix; $y_{k}$ is the estimated output; E is the unit matrix; and λ is the forgetting factor with a value of 0.96.

## 3. Joint Algorithm Estimation

### 3.1. AEKF Estimation of SOC

KF is widely used in linear control systems. The core theory aims to predict the current value based on the estimated value at the previous moment, update the gain and error matrix using the errors of observable variables, and finally, correct the current estimated value to make it close to the true value. The extended Kalman filter, as a derivative algorithm of the Kalman filter, inherits the core of the algorithm of the Kalman filter and can use Taylor expansion to ignore the higher-order terms above the first order and linearize the nonlinear system with the first-order terms, which can be used for the charge state estimation of lithium batteries [41].

The state equation and the observation equation are:(5)$$\{\begin{array}{l} x_{k+1}=f\left(x_{k},u_{k}\right)+\omega_{k}\\ y_{k}=g\left(x_{k},u_{k}\right)+v_{k} \end{array}$$
where $x_{{}_{k}}={\left[\begin{array}{ccc} SOC(k) & U_{{}_{1}}(k) & U_{{}_{2}}(k) \end{array}\right]}^{T}$ is the state variable; $y_{k}=U_{L}(k)$ is the observed variable; $u_{k}$ is the input vector; and $\omega_{k}$ and $v_{k}$ are zero-mean Gaussian white noise. At each moment point, upon linearizing $f\left(x_{k},u_{k}\right)$ and $g\left(x_{k},u_{k}\right)$, we have:(6)$$\{\begin{array}{l} x_{k+1}\approx A_{k}x_{k}+\left[f\left({\hat{x}}_{k},u_{k}\right)-A_{k}{\hat{x}}_{k}\right]+\omega_{k}=A_{k}x_{k}+B_{k}u_{k}+\omega_{k}\\ y_{k}\approx C_{k}x_{k}+\left[g\left({\hat{x}}_{k},u_{k}\right)-C_{k}{\hat{x}}_{k}\right]+v_{k}=C_{k}x_{k}+D_{k}u_{k}+v_{k} \end{array}$$
$$\begin{array}{l} A={\frac{\partial f\left(x_{k},u_{k}\right)}{\partial x_{k}}|}_{x_{k}={\hat{x}}_{k}}=\left[\begin{array}{ccc} 1 & 0 & 0\\ 0 & e^{-\frac{\Delta t}{\tau_{1}}} & 0\\ 0 & 0 & e^{-\frac{\Delta t}{\tau_{2}}} \end{array}\right]\\ B={\left[\begin{array}{ccc} -\frac{\eta \Delta t}{Q_{N}} & R_{1}\left(1-e^{-\frac{\Delta t}{\tau_{1}}}\right) & R_{2}\left(1-e^{-\frac{\Delta t}{\tau_{2}}}\right) \end{array}\right]}^{T}\\ C={\frac{g\left({\hat{x}}_{k},u_{k}\right)}{\partial x_{k}}|}_{x_{k}={\hat{x}}_{k}}=\left[\begin{array}{ccc} \frac{\partial U_{oc}}{\partial SOC} & -1 & -1 \end{array}\right]\\ D=\left[-R_{0}\right] \end{array}$$

Affected by temperature, battery aging, and other factors, there will be real-time changes in interference noise affecting the SOC estimation of a Li-ion battery, but when estimating SOC based on EKF, the process noise covariance matrix Q and observation noise covariance matrix R are often set as constants substituted into the recursive process, which will lead to a decrease in SOC estimation accuracy. To solve the error problem caused by the above reasons, the Sage–Husa adaptive filtering algorithm is introduced to construct the adaptive extended Kalman filtering algorithm (AEKF) so that the noise covariance can be updated adaptively in the recursive process of the algorithm. The adaptive noise covariance equation is:(7)$$\begin{array}{c} H_{k}=\frac{1}{M}\sum_{i=k-M+1}^{k}e_{k}e_{k}^{T}\\ \{\begin{array}{c} R_{k}=H_{k}-C_{k}{\hat{P}}_{k}C_{k}^{T}\\ Q_{k}=K_{k}H_{k}K_{k}^{T} \end{array} \end{array}$$
where M is the dimension of the observation, which is the window size, and is a weighted average of the previous M new interest variance; $e_{k}$ is the new interest matrix, which is the difference between the estimated voltage and the measured voltage; $H_{k}$ is the new interest covariance matrix; P is the error covariance matrix; and K is the Kalman gain. A flow chart of the AEKF-based algorithm is shown in Figure 5.

### 3.2. IGA-BP Algorithm to Compensate for AEKF Errors

#### 3.2.1. BP Neural Network

The BP neural network has a great learning ability because its prediction approach involves adjusting the weights of each layer in turn per the error of the predicted value by continuously correcting the weights between the layers of the model. The input, hidden, and output layers of a BP neural network comprised them [20]. Typically, the input layer is made up of variables that have a strong linear correlation with the output but a weak linear correlation with each other. The intermediate layer’s primary job is to handle the logical connection between the input and output, and the output layer generates the predicted values we are aiming for. The topology based on BPNN is shown in Figure 6.

In Figure 6, $w_{nm}$ and $w_{mk}$ are the connection weights between the different layers of the network. In [20,23], when using BPNN to compensate for EKF estimation errors, the input variables to the BPNN were $\left[\begin{array}{cccccc} SOC(k) & U_{{}_{1}}(k) & U_{2}(k) & K_{1} & K_{2} & K_{3} \end{array}\right]$, and the output was $Err=SOC_{ture}-SOC_{EKF}$. When using the BPNN to compensate for the predictive error of the AEKF, the AEKF-filtered SOC(k), $U_{1}(k)$, $U_{2}(k)$, AEKF gain K, and the error value e of the observed variable are used as input variables, so the input variables are $\left[\begin{array}{ccccccc} SOC(k) & U_{1}(k) & U_{2}(k) & K_{1} & K_{2} & K_{3} & e \end{array}\right]$, and the output layer is the estimation error of AEKF, $Err=SOC_{ture}-SOC_{AEKF}$. It is demonstrated that a three-layer BPNN can fulfill the prediction requirements of nonlinear systems when the number of neurons in the hidden layer is properly determined [27]. The number of neurons in the hidden layer is determined using the empirical formula $y=\sqrt{n+l}+a$ [21], where n and l are the number of neurons in the input and output layers, respectively, and a is a random number between 0 and 10, which is most suitable when y=15 according to several experiments. Therefore, a three-layer BPNN with seven inputs and one output is constructed.

#### 3.2.2. IGA-BP

Due to the unpredictability of the BPNN’s initial parameter selection, local optima are a common problem when employing BPNN for AEKF estimate error compensation. Because of this, the initial weights of the BPNN are optimized in this paper using the stochastic search capability of the genetic algorithm (GA), a stochastic search algorithm invented by John H’s team [42], and simulating the biological genetic and evolutionary elimination mechanism to choose the genetically best individuals via genetic and mutational operations on the offspring. The GA-BP algorithm process is as follows.

Population initialization: Firstly, individuals are coded and arranged in a decimal way, which is advantageous in that the coding and decoding steps can be omitted, making the algorithm process simpler. The gene of each individual is composed of $w_{nm}$ and $w_{mk}$.

Determination of the fitness function: The fitness function is the criterion for the GA model to screen the offspring, which can effectively reflect the gap between individuals in the population. In this paper, we construct the fitness function based on the estimated error sum of squares of AEKF.
(8)$$F={\sum_{i=1}^{N}\left[E\mathrm{rr}-SOC_{BP}\right]}^{2}$$

N is the number of training samples.

Selection operation: The roulette wheel method is usually chosen to determine the next generation, and N individuals in the population require N repetitions. If the fitness value of individual i is $F_{i}$, the probability that individual i is left behind at each selection is:(9)$$P_{si}=\frac{\frac{k}{F_{i}}}{\sum_{j=1}^{N}\frac{k}{F_{j}}}$$
where k is the coefficient.

Crossover operation: Since individuals are coded with real numbers, the real number crossover method is chosen, and chromosomes l, $a_{l}$, and chromosomes k, $a_{k}$, are crossed at position j as follows:(10)$$\{\begin{array}{c} a_{lj}=a_{lj}(1-b)+a_{kj}b\\ a_{kj}=a_{kj}(1-b)+a_{lj}b \end{array}$$
where b is a random number, $b\in \left[\begin{array}{cc} 0, & 1 \end{array}\right]$.

Mutation operation: the j gene of individual i is selected for mutation, and the mutation mode is shown in Formula (10).
(11)$$a_{ij}=\{\begin{array}{c} a_{ij}+(a_{ij}-a_{\max})*f(g)\;\;r\geq 0.5\\ a_{ij}+(a_{\min}-a_{ij})*f(g)\;\;r<0.5 \end{array}$$
where $a_{\max}$ and $a_{\min}$ are the upper and lower thresholds of gene $a_{ij}$, f(g)=r(1−gGmax), $r\in \left[\begin{array}{cc} 0, & 1 \end{array}\right]$. g is the number of current iterations, and is $G_{\max}$ the maximum number of iterations.

After several iterations of the genetic algorithm, the number of good individuals will grow exponentially. Since the population size is limited, the dramatic increase in good individuals will lead to the rapid disappearance of other individuals. This is similar to the law of “survival of the fittest” in nature, but a population with mostly genetically similar good individuals will lead to increased inbreeding, and the algorithm will easily fall into the local extreme value trap. If the crossover probability is too small, the algorithm will be slowed down, and if it is too large, highly adapted individuals will be destroyed quickly. If the mutation probability is too small, it will be difficult to generate new individuals, and if it is too large, the algorithm will be meaningless. In addition, this paper establishes a fitness function based on the prediction value error, so the lower the value of fitness, the better the individual is. Therefore, the cosine function is introduced to design the adaptive crossover probability and variation probability, and thus, improve population diversity.
(12)$$p_{c}=\{\begin{array}{ll} p_{c1}\cos\left(\frac{\pi }{2}\frac{F_{avg}-F}{F_{avg}-F_{\min}}\right) & F⩽F_{avg}\\ p_{c2}\left(1-\cos\left(\frac{\pi }{2}\frac{F-F_{avg}}{F_{\max}-F_{avg}}\right)\right) & F>F_{avg} \end{array}$$
where $p_{c}$ is the crossover probability, $p_{c1}$ and $p_{c2}$ are the maximum crossover probabilities of high-quality and low-quality individuals, respectively, $F_{\max}$, $F_{\min}$, and $F_{avg}$ are the maximum, minimum, and average fitness values in the population, respectively, and F is the fitness value of the individuals to be crossed.
(13)$$p_{m}=\{\begin{array}{ll} p_{m1}+k_{1}\cos\left(\frac{\pi }{2}\frac{F_{avg}-F}{F_{avg}-F_{\min}}\right) & F⩽F_{avg}\\ p_{m2}+k_{2}\left(1-\cos\left(\frac{\pi }{2}\frac{F-F_{avg}}{F_{\max}-F_{avg}}\right)\right) & F>F_{avg} \end{array}$$
where $p_{m}$ is the probability of variation, $p_{m1}$ and $p_{m2}$ are the minimum variation rates of excellent and inferior individuals, respectively, and $k_{1}$ and $k_{2}$ are coefficients.

At the same time, the simulated annealing algorithm (SA) was introduced in the GA [43]. SA is based on the fact that the free motion entropy of the molecules of a substance increases at high temperatures, leading to high activity, causing them to rearrange randomly and slowly stabilizing the molecules during the cooling process. The SA algorithm accepts non-optimal solutions with a certain probability according to the Metropolis criterion, which, to a certain extent, compensates for the easy maturation of GA in the early stage and stagnation in the later stage.

According to the Metropolis criterion, the optimal individuals in the old and new populations are determined. The probability of accepting a new individual is:(14)$$p=\{\begin{array}{cc} 1 & F(c)\leq F_{\min}\\ e^{\frac{F_{\min}-F(c)}{T}} & F(c)>F_{\min} \end{array}$$

If the fitness value of the optimal individual in the new population is lower than that of the old individual, the individual is accepted as the new optimal value; otherwise, the acceptance probability of the new individual is calculated according to the formula, while a random number between 0 and 1 is generated, and if the acceptance probability of the new individual is greater than the random number, the new individual is accepted; otherwise, it is not accepted. At the end of each genetic operation, annealing is required. $T=T_{0}*k$, $T_{0}$ is the initial temperature, k is the annealing coefficient, and k∈[0.9,1]. When the algorithm starts, T is larger and the GA is more likely to accept the inferior solution in the new population. The value gradually decreases with the increase in population generations, and the probability of accepting the inferior new solution becomes lower and lower, and when the value of T tends to 0, it basically no longer accepts any deteriorating new solutions, and can only accept a new solution with a state lower than the current solution; thus, this is the key to preventing the GA from prematurely maturing and jumping out of the local extremes. A flow chart of the improved genetic algorithm for optimizing the BPNN is shown in Figure 7.

### 3.3. SOC Estimation Based on the Joint IGA-BP-AEKF Algorithm

The equivalent circuit method is computationally simple and can reflect the operating characteristics of the battery based on the external characteristics of the battery, and the data-driven method can be trained using a large amount of data so as to find the mapping relationship between the input and output. The fusion algorithm combines the advantages of both, and the optimized BPNN is trained offline. The trained BPNN compensates for the SOC estimation error of the AEKF so that the SOC estimation result is closer to the real value. The specific flow of the fusion algorithm is shown in Figure 8.

From the figure, we can see that the whole prediction model is divided into three parts: parameter identification, AEKF prediction of SOC, and IGA-BP compensation.

Step 1: BPNN training. A 3-layer BPNN with seven inputs and one output is established, and the implied layer neuron number is 15. The voltage and current data under UDDS conditions are collected, the AEKF is used to estimate the SOC under UDDS conditions, and the $K_{k}$, $e_{k}$, and $x_{k}^{+}$ generated about the AEKF estimation process are saved. Figure 9 shows the SOC estimation under UDDS conditions using the AEKF.

The Kalman filter gain $K_{k}$, the end voltage observation error $e_{k}$, and the state prediction value $x_{k}^{+}$ are used as the input of the IGA-optimized BPNN, and Err is used as the output. The learning rate is set to 0.001, and the expectation value is 1 × 10^−5^. However, the data need to be normalized according to Equation (15) before input to avoid the different ranges of the input variables that lead to network training Failure.
(15)$$\bar{x}=\frac{x_{i}-x_{\min}}{x_{\max}-x_{\min}}$$
where: $x_{\max}$ and $x_{\min}$ are the maximum and minimum values of the original data. In total, 80% of the data are chosen as the training sample and 20% as the test sample. When the predicted desirable value is obtained, the intended BPNN training is proven to be accomplished [21].

Step 2: Joint estimation of parameter identification and AEKF algorithm. As can be seen from the figure, based on the real-time battery measurement data, FFRLS is used to realize the online identification of the battery model parameters, and then, the AEKF is used to complete the estimation of SOC and the update of $K_{k}$, $e_{k}$, and $x_{k}^{+}$ based on the obtained model parameters, and at the same time, the updated open-circuit voltage value is obtained from the OCV-SOC curve based on the SOC value at this time. This is substituted into the FFRLS algorithm to complete the two-step identification of the model parameters, and therefore, to achieve the joint estimation of model parameters and the AEKF algorithm.

Step 3: Joint estimation of BPNN and AEKF. $K_{k}$, $e_{k}$, $x_{k}^{+}$, and Err are input into the successfully trained BPNN. Finally, the sum of the optimal estimate after AEKF filtering and the output of the BPNN are obtained as the optimal estimate of the fusion algorithm.

## 4. Experimental Verification Analysis

In actual operation, the battery cannot be guaranteed to work at a constant current all the time, and the current output of the battery is complex and variable. This study was conducted in a MATLAB working environment to simulate the battery operation mode for the experiments, assuming that the battery was fully charged at the initial moment and setting the initial value of SOC to 1. The validity of the proposed estimation method is demonstrated by the complex operating condition FUDS. The current and voltage data of FUDS operating conditions are shown in Figure 10.

### 4.1. Validation of the Parameter Identification Effect

In the model identification effect validation experiments, the FFRLS algorithm is used to estimate the end voltage values using the data collected in the laboratory. Figure 11(left) shows the comparison between the estimated end voltage based on FFRLS and the measured voltage under the second-order RC equivalent circuit model. The estimated voltage basically matches with the measured voltage, which verifies that the second-order equivalent model has high stability and that the FFELS identification effect is good. Figure 11(right) shows that the voltage estimation error tends to increase gradually as the charging and discharging proceeds, which is caused by the more intense internal chemical reaction that occurs when the lithium battery is discharged to the cutoff voltage state, and the maximum absolute value of the end voltage error based on the FFRLS algorithm is calculated to be 0.035. The results show that the FFRLS can identify the model parameters well under the second-order RC model for subsequent SOC estimation use.

### 4.2. AEKF Algorithm Validation

In order to verify that the SOC estimation accuracy of the EKF algorithm is effectively improved after the introduction of the adaptive algorithm, the EKF and AEKF algorithms are simulated and validated using FUDS operating data as a comparison object. Except for the introduction of the adaptive algorithm, the initial parameters of the two algorithms are the same. The simulation results are shown in Figure 12(left). The SOC estimates under both algorithms follow the true values well, but the estimates of the AEKF algorithm are closer to the reference values. The error results are shown in Figure 12(right); the SOC error estimated by the AEKF is smaller than that of the EKF, but the SOC estimation error curves of both the EKF and AEKF models show large fluctuations when the current changes abruptly, which is caused by the large errors in the model parameters identified by FFRLS when the current changes abruptly, which coincides with the identification voltage error curve in Figure 11(right). To further examine the model prediction performance, the root mean square error (RMSE) and mean absolute error (MAE) are used to evaluate the two models. Table 1 shows that the maximum error of EKF is 0.0576 and that of AEKF is 0.0432. The RMSE and MAE values of AEKF are lower than those of EKF, so the noise reduction capability of the adaptive algorithm is better and the SOC estimation accuracy is significantly improved.
(16)$$\mathrm{RMSE}=\sqrt{\frac{1}{N}\sum_{i=1}^{N}{\left(Y_{i}-f_{i}\right)}^{2}}$$
(17)$$\mathrm{MAE}=\frac{1}{N}\sum_{i=1}^{N}\left|Y_{i}-f_{i}\right|$$
where N is the total amount of data, $Y_{i}$ the true value, and $f_{i}$ is the predicted value.

### 4.3. IGA-BP-AEKF Algorithm Validation

The SOC estimation results of the proposed joint algorithm and the unimproved algorithm were compared with all initial values being equal, and the experimental results are shown in figure. From Figure 13(left), it can be seen that the prediction trends of the three algorithms are basically consistent with the true values, but the SOC curve fit of the IGA-BP-AEKF algorithm is closer to the reference value, reflecting its strong tracking ability. From Figure 13(right), we can see that the maximum error of AEKF is significantly reduced after the BPNN compensation, and the BPNN compensation of AEKF after IGA optimization has the best effect. In the SOC < 0.5 stage, unlike the other two algorithms, the IGA-BP-AEKF model does not have a large abrupt change at a certain moment, indicating that IGA enhances the global optimization-seeking ability in the genetic process, has more advantages in searching the optimal initial parameters of BPNN, and has strong anti-interference ability, which is consistent with the performance of the error curve. From Table 2, we know that IGA-BP-AEKF has the smallest data values under all three evaluation criteria, which indicates that the improved joint algorithm has better stability in estimating SOC.

## 5. Conclusions

In view of the shortage of related research on the SOC estimation of lithium-ion batteries, a novel fusion method for SOC estimation based on improved Genetic Algorithm BP and an adaptive extended Kalman filter is proposed in this paper. The experimental results indicate that the proposed IGA-BP-EKF algorithm is superior to some other related ones. The maximum absolute value of the terminal voltage error based on the FFRLS algorithm for online parameter identification in the second-order RC model is calculated to be 0.035, which is good enough for subsequent accurate SOC estimation. The experimental results under FUDS working conditions show that the maximum error of the IGA-BP-EKF algorithm is 0.0119, MAE is 0.0083, and RMSE is 0.0088, which greatly improves the accuracy of the SOC estimation of lithium-ion batteries compared with BP-AEKF and GA-BP-AEKF. In future work, the effects of temperature and other factors should be considered on the SOC estimation of Li-ion batteries, and experimental operating condition data should be enriched and improved.

## Figures and Tables

**Figure 1 sensors-23-05457-f001:**
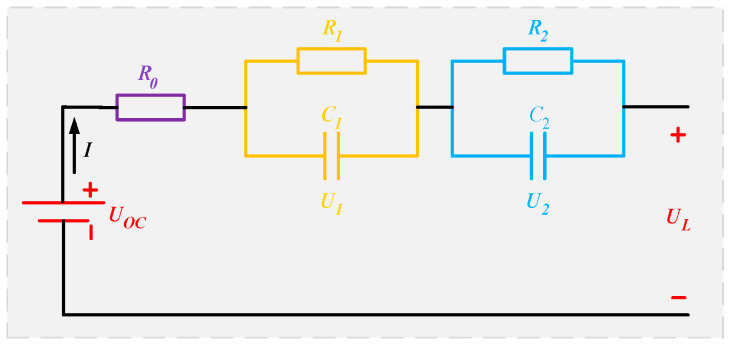
Second-order RC model.

**Figure 2 sensors-23-05457-f002:**
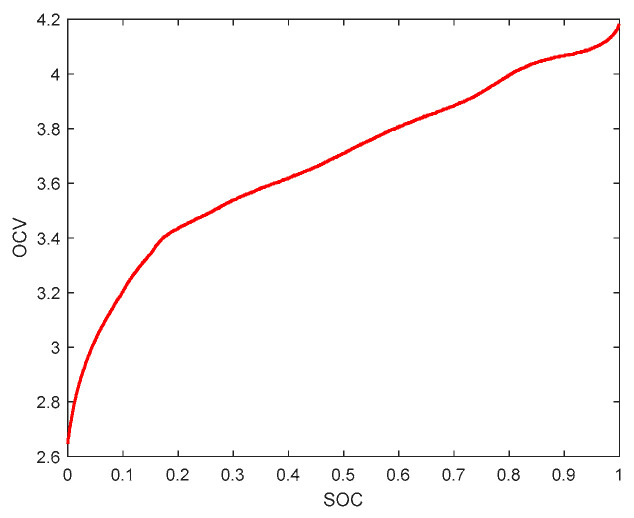
OCV-SOC curve.

**Figure 3 sensors-23-05457-f003:**
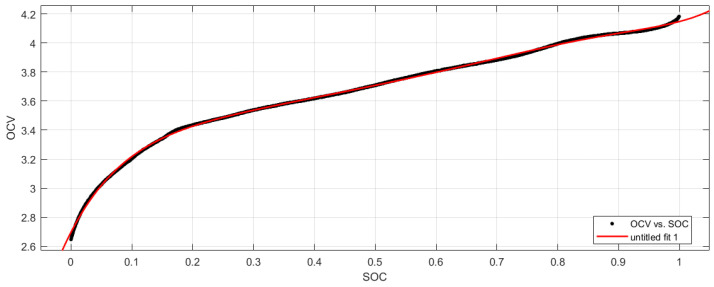
OCV-SOC curve fitting results.

**Figure 4 sensors-23-05457-f004:**
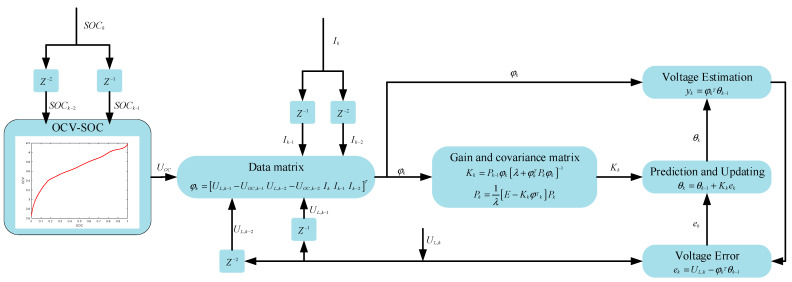
Model identification flow chart based on FFRLS.

**Figure 5 sensors-23-05457-f005:**
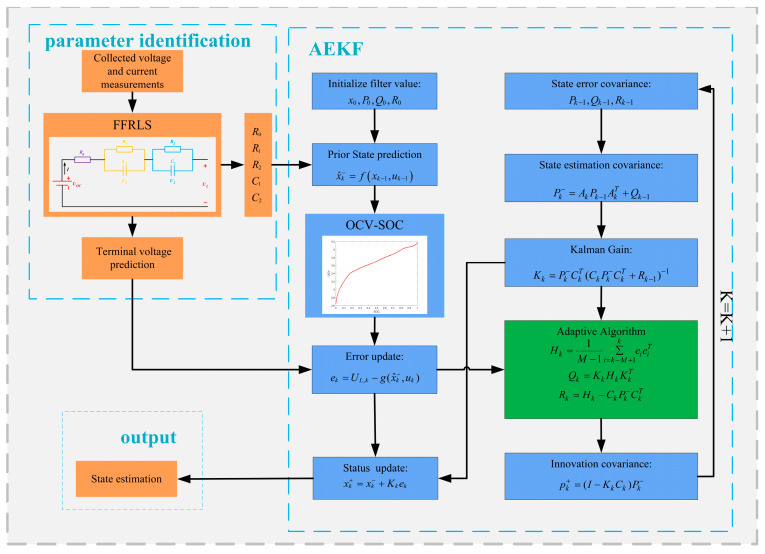
Flow chart of SOC estimation based on AEKF.

**Figure 6 sensors-23-05457-f006:**
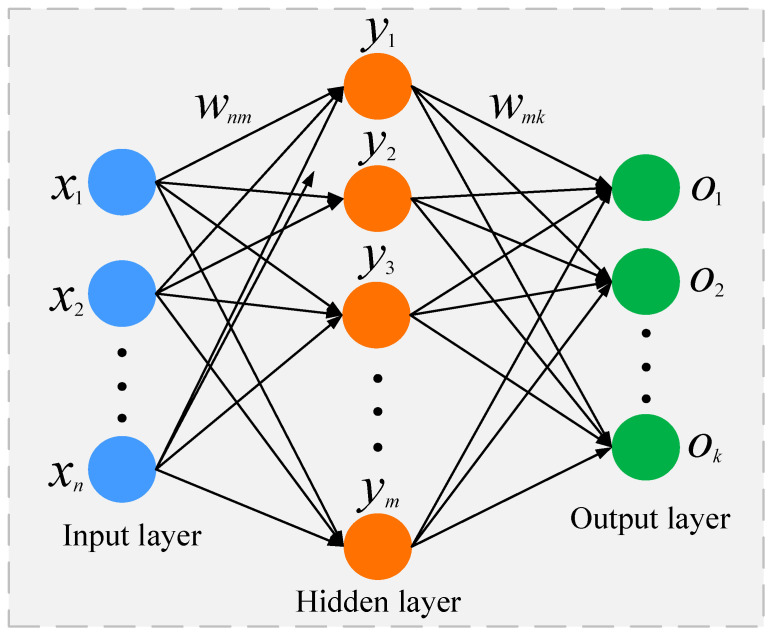
BPNN structure diagram.

**Figure 7 sensors-23-05457-f007:**
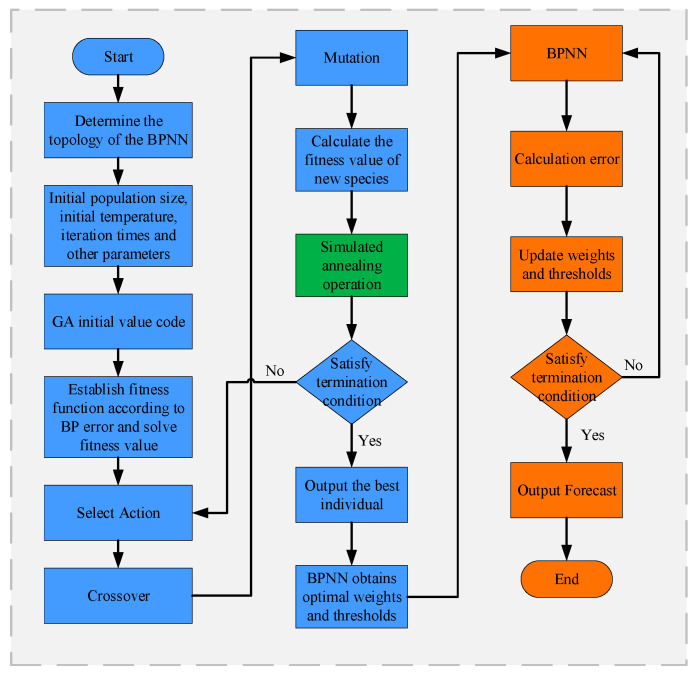
IGA-BP flow chart.

**Figure 8 sensors-23-05457-f008:**
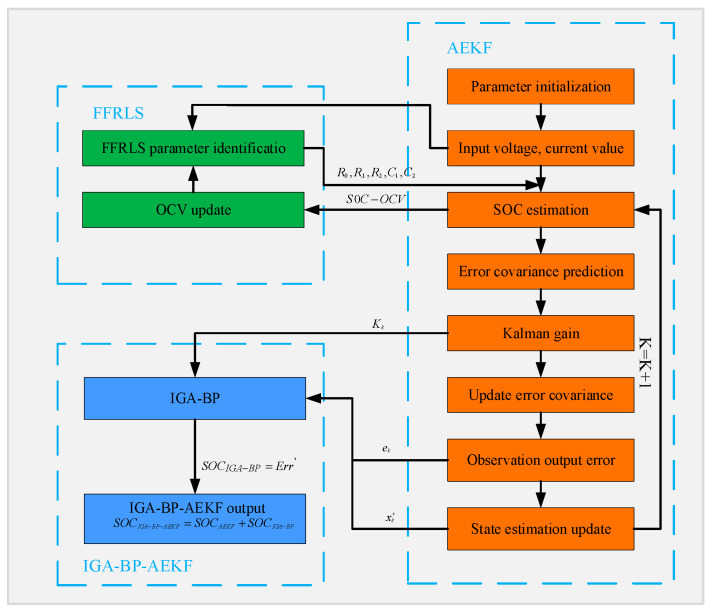
SOC estimation based on IGA-BP-AEKF.

**Figure 9 sensors-23-05457-f009:**
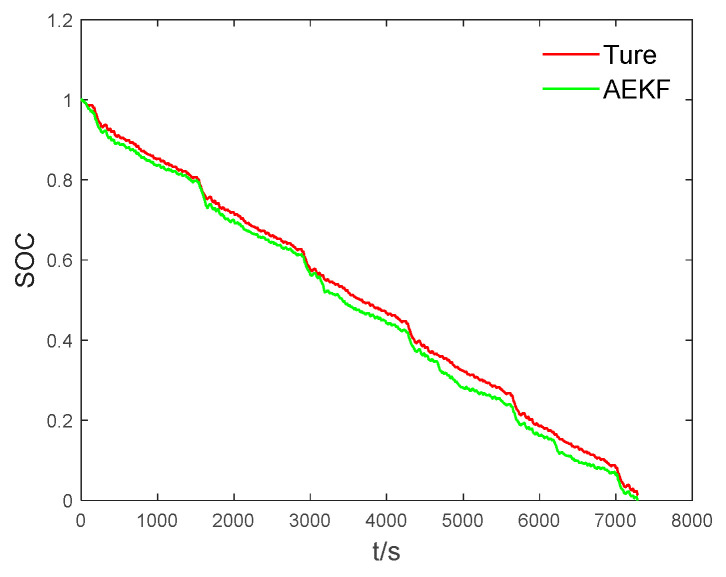
AEKF estimated SOC diagram under UDDS working conditions.

**Figure 10 sensors-23-05457-f010:**
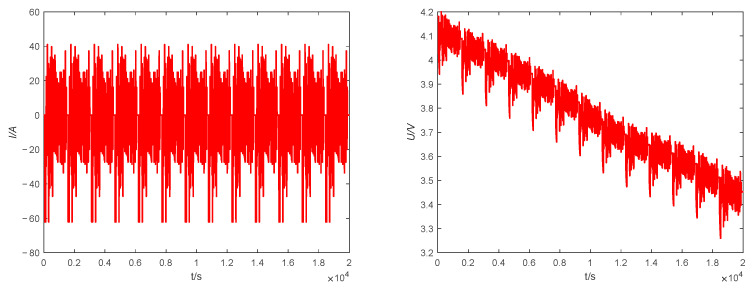
FUDS data. **Left**: current, **right**: voltage.

**Figure 11 sensors-23-05457-f011:**
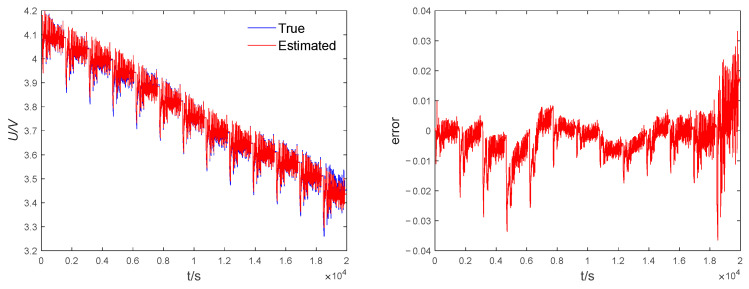
FFRLS identification results. **Left**: voltage comparison, **right**: voltage error.

**Figure 12 sensors-23-05457-f012:**
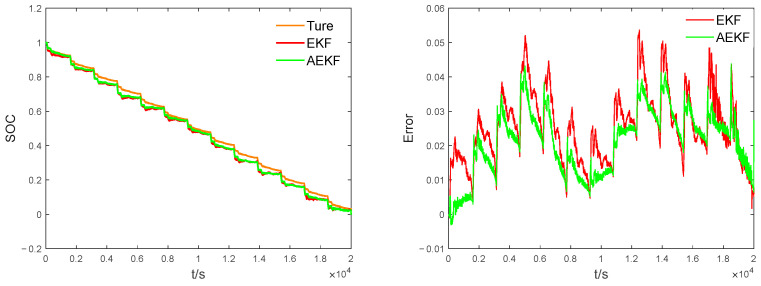
Comparison of the estimated SOC of EKF and AEKF. **Left**: SOC comparison results, **right**: comparison of estimation errors of SOC.

**Figure 13 sensors-23-05457-f013:**
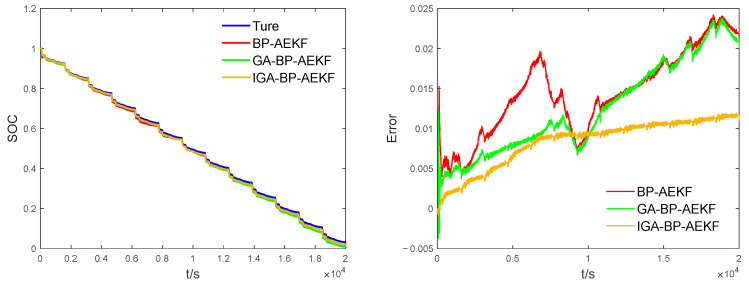
Estimation results of SOC under different joint algorithms. **Left**: comparison of SOC estimates, **right**: comparison of estimation errors of SOC.

**Table 1 sensors-23-05457-t001:** Comparison of SOC estimation errors between EKF and AEKF for UDDS operating conditions.

Estimation Method	Maximum Err	MAE	RMSE
EKF	0.0576	0.0254	0.0274
AEKF	0.0438	0.0212	0.0232

**Table 2 sensors-23-05457-t002:** Comparison of SOC estimation errors of different joint algorithms under UDDS conditions.

Estimation Method	Maximum Err	MAE	RMSE
BP-AEKF	0.0242	0.0144	0.0153
GA-BP-AEKF	0.0237	0.0124	0.0137
IGA-BP-AEKF	0.0119	0.0083	0.0088

## Data Availability

The data presented in this study are available upon request from the corresponding author.

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
