# Peer review of "A Novel Fusion Method for State-of-Charge Estimation of Lithium-Ion Batteries Based on Improved Genetic Algorithm BP and Adaptive Extended Kalman Filter"

_sensors, 2023, doi:10.3390/s23125457_

Round 1

Reviewer 1 Report

Manuscript entitled "A Novel Fusion Method for....." reports on  improving the accuracy of SOC estimation through a novel fusion method IGA-BP-AEKF . Manuscript is well written and it can be accepted in the present format after incorporating the following minor change.

Figure qulaity must be improved. For example, Figure 4 and 5 can be plotted in a much better way

Author Response

On behalf of my co-authors, we appreciate the editor and the reviewer very much for the positive and constructive comments and suggestions on our manuscript.

Manuscript ID: sensors- 2286583,

Title: A Novel Fusion Method for State-of-Charge Estimation of Lithium-ion Batteries based on Improved Genetic Algrithm -BP and Adaptive Extended Kalman Filter.

We carefully studied the comments and found the corresponding questions in the manuscript. We have tried our best to revise our manuscript according to the comments. Attached is a revised version that we would like to submit for your consideration. We would like to express our great appreciation to editor and reviewers for comments on our paper. Looking forward to hearing from you.

Reviewer 2 Report

This paper proposes a novel method for vehicle battery state estimation. In general, the paper is interesting and has potential. However, there are still a few problems that need to be carefully addressed. More specifically,

1. Please highlight the contribution of your work at the end of the introduction.

2. Currently, there are a lot of Kalman filters and their variants applications to vehicle state estimation with multi-information fusion. Thus, some of the work should be discussed: autonomous vehicle kinematics and dynamics synthesis for sideslip angle estimation based on consensus kalman filter; yolov5-tassel: detecting tassels in rgb uav imagery with improved yolov5 based on transfer learning; vision-aided intelligent vehicle sideslip angle estimation based on a dynamic model; estimation on imu yaw misalignment by fusing information of automotive onboard sensors; automated vehicle sideslip angle estimation considering signal measurement characteristic; improved vehicle localization using on-board sensors and vehicle lateral velocity.

3. Please correct the equation number carefully.

4. The result analysis should be in-depth.

5. Future work should be discussed at the end of the paper.

Author Response

(The authors gave the same response as above.)

Reviewer 3 Report

Here is presented a SOC estimation of Lithium-ion batteries, based on improved Genetic Algorithm- BPNN and adaptive extended Kalman filter. Such data driven methods are interesting and many works on the same problem exist. Battery safety is important problem and its algorymization is emerging.

I have noticed some weak points in the manuscript. They are:

All equations are wrongly numbered.

In section 2.1. Battery Modeling - Not clear where SOC comes from in equation (2)?

In section 2.2. OCV-SOC acquisition – please define battery parameters use for the modeling. Please explain how this model is effected by temperature or degradation. Predicting SOC by voltage only is useless.

How did you estimate battery SOC where it is on the input of BPNN? Where did you get the initial SOC?

In Figure 8 – IGA does not provide any data. It must be for BPNN training only. Why it is in the main processing flow?

Training and testing are made by the same type of simplified battery model. Could you consider testing on benchmark actual battery measured V-I data?

In conclusion, presented method is not well described. Its applicability, outside pure computation exercise, is questionable. 

Author Response

(The authors gave the same response as above.)

Round 2

Reviewer 3 Report

Changes are reasonable. Paper is acceptable for publication.

Look again equation numbering and symbols indexation. x and y are used for KF, but also used in BPNN, same issue for weight coefficients in KF and BPNN, symbols can be different.